# Recognition of Pharmacological Bi-Heterocyclic Compounds by Using Terahertz Time Domain Spectroscopy and Chemometrics

**DOI:** 10.3390/s19153349

**Published:** 2019-07-30

**Authors:** Maciej Roman Nowak, Rafał Zdunek, Edward Pliński, Piotr Świątek, Małgorzata Strzelecka, Wiesław Malinka, Stanisława Plińska

**Affiliations:** 1Faculty of Electronics, Wroclaw University of Science and Technology, Wybrzeze Wyspianskiego 27, 50-370 Wroclaw, Poland; 2Department of Chemistry of Drugs, Faculty of Pharmacy, Wroclaw Medical University, Wroclaw Ludwika Pasteura 1, 50-367 Wroclaw, Poland; 3Department of Inorganic Chemistry, Faculty of Pharmacy, Wroclaw Medical University, Wroclaw Ludwika Pasteura 1, 50-367 Wroclaw, Poland

**Keywords:** (THz-TDS) terahertz time domain spectroscopy, (KSVM) kernel support vector machine, (kNN) k-nearest neighbour, bi-heterocyclic compounds, chemometrics

## Abstract

In this study, we presented the concept and implementation of a fully functional system for the recognition of bi-heterocyclic compounds. We have conducted research into the application of machine learning methods to correctly recognize compounds based on THz spectra, and we have described the process of selecting optimal parameters for the kernel support vector machine (KSVM) with an additional ‘unknown’ class. The chemical compounds used in the study contain a target molecule, used in pharmacy to combat inflammatory states formed in living organisms. Ready-made medical products with similar properties are commonly referred to as non-steroidal anti-inflammatory drugs (NSAIDs) once authorised on the pharmaceutical market. It was crucial to clearly determine whether the tested sample is a chemical compound known to researchers or is a completely new structure which should be additionally tested using other spectrometric methods. Our approach allows us to achieve 100% accuracy of the classification of the tested chemical compounds in the time of several milliseconds counted for 30 samples of the test set. It fits perfectly into the concept of rapid recognition of bi-heterocyclic compounds without the need to analyse the percentage composition of compound components, assuming that the sample is classified in a known group. The method allows us to minimize testing costs and significant reduction of the time of analysis.

## 1. Introduction

A heterocyclic compound is where the atoms form a ring, with at least one non-carbon atom in a single cyclic system. Elements such as oxygen, nitrogen and sulphur appear most often in the ring [1]. Heterocyclic compounds are common in the life of every human being [2,3]. The best examples are amino acids (histamine, serotonin, melatonin, hydroxyproline), vitamins (e.g., folic acid or vitamin M, biotin = vitamin H, thiamine = vitamin B1, riboflavin = B2, niacin = B3, pyridoxine = B6, cyanocobalamin = B12), coenzymes (flavin adenine dinucleotide (FAD), reduced form FAD (FADH2), nicotinamide adenine dinucleotide phosphate (NADP+)), dyes (hem/haemoglobin, chlorophyll a), nucleic acids (adenine, guanine, cytosine, thymine, uracil) directly related to deoxyribonucleic acids (DNA) and ribonucleic acids (RNA)), insecticides (e.g., fipronil), toxins (ergotamine, divicine, muscarine) and natural and synthetic flavourings [4]. Analysing these examples, it can be concluded that a large part of the applications find their place in medicine. It is estimated that the group of compounds containing at least one heterocyclic ring constitutes over 70% of all pharmaceuticals [5]. Heterocyclic rings are extremely important structural elements of numerous pharmaceutical preparations currently available in pharmacies and constitute the basis for searching for new biologically active substances [6]. An excellent example is pyrimidine and its derivatives used to treat Parkinson’s disease, neuropathy, bacteria, viruses, fungi, cancer and hypertension [7]. Drugs based on the so-called quinazoline skeleton have similar properties. Quinazoline is a structure formed by combining two aromatic hexavalent rings: benzene and pyrimidine, which means a bi-heterocyclic structure. Some quinazoline derivatives can additionally treat inflammatory states, allergies, autoimmune diseases, eye diseases and even lower blood glucose levels [8]. A greater variety of heterocyclic structures, methods of their production and detailed applications are described in numerous specialist books related to pharmacy and medical chemistry [1,8,9,10].

### 1.1. Test Methods Used to Diagnose Drugs

The techniques used for drug diagnosis usually include Raman spectroscopy [11,12] and near infrared spectroscopy (NIR) [13,14]. Both techniques allow at-line, online and in-line measurements, which makes them very practical and useful in pharmacy and medical chemistry [15]. Liquid chromatography (LC) [16], X-ray diffraction (XRD) [17], and mass spectroscopy [18,19] are also commonly used in chemistry to examine the percentage composition of elements. Nuclear quadrupole resonance spectroscopy (NQR) plays a particularly important role [20]. The radio frequency (RF) method is used to identify counterfeit medicines in packages [21]. Drugs, especially recreational drugs, can also be diagnosed using neutron scattering techniques, in which the densities of H, C, N, O atoms in the substance are determined [22]. In the case of prohibited drugs such as methacvalone (Mandrax), cocaine, LSD, heroin, and morphine, the diagnosis is based on the detection of high concentrations of hydrogen and carbon and low concentrations of nitrogen and oxygen [23].

The last technique mentioned by us is THz spectroscopy, which is the basis of our research. When synthesizing new molecules (drug candidates), other large and complex molecules (molecular tails) are attached to the basic molecule (core). In the THz band, we observe how various tails affect the resulting spectrum. In the IR band and especially NIR, one can observe vibrations between particular atoms, which can often disturb spectral observations and even make the interpretation of the results more difficult. Only the examination of intermolecular vibrations makes it possible to distinguish individual synthesized molecules, and this is the main advantage of THz spectroscopy. The best example is the work that presents the results on the usage of THz and IR spectra for the analysis of three derivatives of the 1,4-naphthoquinone compound (plumbagin, juglone i menadione) used in the traditional Chinese medicine. The IR spectra for all three compounds were very similar and required a detailed analysis of all functional groups, while the THz range allowed for recognizing the compounds (molecular fingerprints) unambiguously. Many scientific publications have been devoted to the use of terahertz techniques for the detection and identification of prohibited drugs [24,25]. Using the Terahertz time spectroscopy (THz-TDS) system with imaging and knowledge of the location of absorption peaks of drugs, it is possible to detect prohibited psychoactive substances [26,27]. The terahertz parametric oscillator (TPO) [28] and injection-seeded terahertz parametric generation (is-TPG) methods [29] deserve special recognition. They make it possible to detect, among other things, illegal drugs and many reagents, as well as the inspection of the pharmaceutical production process with a dynamic range of about four to more than eight orders of magnitude. This allows for imaging of objects that are thicker than a thin envelope. Another application of terahertz pulsed imaging (TPI) is the study of drug coatings by means of terahertz pulse imaging (TPI) allowing for three-dimensional characterization of the tablet structure [30,31]. Coating with a thin layer of polymer is used, among others, to improve the stability of parameters, facilitate swallowing and change the drug release time [32]. This method can be successfully used to inspect the top layer thickness of tablets on the production line in real time [33,34]. Monitoring of drugs during production and storage is also possible by examining the crystalline structure of samples with the use of terahertz radiation [35]. This is particularly important in the event of changes in environmental conditions (temperature, humidity, etc.) in the production process and may affect the final composition and action of the drug. Research on polymorphic forms of drugs and descriptions of terahertz techniques can be found in numerous publications [36,37]. The terahertz technique may help identify molecular structures of drugs with subtle differences in functional groups [38]. The authors point to the high effectiveness of the diagnosis of some diabetic drug pharmacophores in relation to density functional theory (DFT) limitations. Research was also conducted on the diagnosis of amino acids in food supplements [39]. It should be noted that terahertz techniques can be used as autonomous or in many applications they support other techniques. The use of mixed technique, a combination of Raman spectroscopy and TPI imaging spectroscopy [40] allowed the development of a method of drug release time prediction. Detailed applications of THz are described in review publications [41,42,43,44] and books [45,46,47]. The THz-TDS system has also been used in [48] for predicting physicochemical properties of the drugs based on piroxicam. In this work, the Partial Least Squares (PLS) -based predictive model was built to quantify the relationship between the spectra and the melting point, which can guide the selection of early drug candidates.

### 1.2. Drug Detection Using Terahertz Spectroscopy Combined with Support Vector Machine (Svm)

The recognition of drugs on the basis of their terahertz spectra requires the use of an appropriate mathematical apparatus. Statistical methods are useful, especially the algorithms that are based on pattern recognition methods. Numerous papers were devoted to the use of the SVM algorithm for drug analysis. The basic role of SVM is to determine whether a given structure is a pharmaceutical drug or not [49,50], and, if such a structure can be included in the group of pharmaceutical drugs, what its biological activity, chemical reactivity or toxicity will be [51,52,53]. Prediction of physical and/or biological properties of a given drug is usually done on the basis of a certain number of descriptors, which are appropriately tabulated (e.g., Ghose Crippen descriptors, topological pharmacophore (CATS) and others). Reverse prediction can also be made, i.e., knowing the appropriate relation between the structure of the drug with e.g., its activity, the structure of the tested drug is recognized [49,54,55].

There are only a few studies on the use of SVM and THz spectra for the identification of drugs or candidates [56,57] of drugs other than the prohibited. In [56], the THz spectra were used for recognition of traditional Chinese medicaments with an artificial neural network (ANN) and Kernel support vector machine (KSVM) with untrained linear, polynomial and Gaussian radial basis function (GRBF) kernels. KSVM outperformed ANN for all the medicaments at the constant parameters of the kernels. In our study, we applied KSVM with an adaptive kernel, which is much more suitable for practical applications. It should be also mentioned that the initial results on using KSVM for recognition of bi-heterocyclic compounds (BHCs) were presented in the conference correspondence [57] but also with constant parameters in a kernel.

In this study, we focus on the classification of 13 bi-heterocyclic compounds that have not yet entered production but are at the ‘in vitro’ testing stage. THz-TDS and self-optimizing SVM combined with the *k*-Nearest Neighbour (kNN) algorithm were used to identify the samples under investigation. The exact application of the algorithms is described in Section 3.

The remainder of this paper is organized as follows: Section 2 presents the experimental setup and a short description of the analyzed bi-heterocyclic compounds. The statistical tools are described in Section 3. The classification results are presented in Section 4. Finally, the conclusions are drawn in Section 5.

## 2. Experimental Study

A system for the recognition of pharmacological bi-heterocyclic compounds is shown in Figure 1. Its main components are a terahertz spectrometer (THz-TDS) combined with a motorized rotary sample holder (MRSH) and a computer with MATLAB software (version 2016a, 64-bit, MathWorks, Natick, MA, USA).

### 2.1. TDS-THz Spectrometer

Spectral measurements of samples were performed using the classical terahertz spectrometer THz-TDS. Femtosecond laser was characterized by an average power of 200 mW, pulse duration of 8.5 fs and pulse repetition frequency of 100 MHz. The 780 nm laser output beam is divided into two beams—excitation (EB) and sampling (PB)—by means of a light-dividing cube (BS). The first one goes to the transmitting antenna (Tx), and the second to the receiving antenna (Rx). In both cases, the same standard LT-GaAs photoconductive antennae have been used, resulting in a useful bandwidth of 2.1 THz. Both beams (EB and PB) are guided by means of mirrors and focused on the gap between antenna dipoles by means of microscopic lenses (MO). The terahertz beam path was made by means of four parabolic off-axis mirrors (PM1-4). Between PM2 and PM3, there is a rotating, numerically controlled holder for 16 test samples. The rotation is realized in such a way that, after registering the spectrum of one compound, the handle is rotated by 22.5${}^{\circ }$ to the position of the next sample. Connections of signal generator, lock-in amplifier and delay line (DS) (retroprism mounted on electric linear drive) shown in Figure 1 recreate the pulse on Rx. The range of the band for the studied bi-heterocyclic compounds was limited to 10–70 cm${}^{-1}$.

### 2.2. Method of Manufacture of the Samples

A single sample is a crystalline form of a bi-heterocyclic compound together with a pure form of polyethylene (PE) formed as a pill (pellet) with a diameter of 13 mm. The weight ratio of pure compound to PE is 40 mg to 360 mg, which corresponds to a percentage ratio of 10% to 90%. The reference sample is a PE pill that has been reduced to 360 mg in weight. A hydraulic press of two tonnes for two minutes is used to make pellets.

### 2.3. Chemical Structure of Bi-Heterocyclic Compounds

The chemical structures of all compounds tested by us, labelled as BHC1-13, are presented in Figure 2. They belong to a group of bi-heterocycles with a common core based on isothiazolpyridine derivatives, more precisely 4,6-dimethyl-2,3-dihydroisothiazolo[5,4-b]pyridin-3-on (Figure 2, last item, in blue). These are compounds inhibiting cyclooxygenases COX-1 and COX-2 [58,59] and are still in the ‘in vitro’ testing phase. Drugs that have similar properties but are registered and are commonly found in the pharmaceutical market are referred to as non-steroidal anti-inflammatory drugs (NSAIDs). Terahertz spectra of visible structures are presented in the further part of the study (Section 3.1, Figure 3).

## 3. Statistical Analysis

Terahertz spectra of bi-heterocyclic compounds contain regular structures that can be analyzed with advanced machine learning methods. The aim is to design a recognition method that can uniquely recognize bi-heterocyclic compounds based on their observed THz spectra. Our approach consists of two stages. First, relevant features are extracted from a set of observed spectra using principal component analysis (PCA) that belongs to a family of unsupervised machine learning methods. In the other stage, supervised learning methods are used to classify the features. We observed that the measured spectral samples, projected onto the space spanned by the PCA features, form regular clusters that might be separated by a nonlinear hypersurface. This motivates the usage of a nonlinear support vector machine (SVM) classifier with an adequately specified kernel function. To maximize the margin between the samples projected onto the nonlinear feature space, we propose to learn the parameters of the kernel function from training data using the cross-validation (CV) technique and the sequential quadratic programming (SQP) method [60] to minimize the misclassification error function. The concept of the proposed learning algorithm is presented below.

### 3.1. PCA

To design a supervised learning recognition algorithm, the observed THz spectra are partitioned into training and testing samples, e.g., according to the *n*-fold CV rule. We assume that the testing spectra can be classified into *C* disjoint classes. The examples of observed spectra are shown in Figure 3.

Spectral resolution of observed spectra is usually high, and each spectrum contains a lot of redundant information. Reduction of redundancy can therefore be performed with PCA. It extracts *J* orthogonal features that trace the directions in the *I*-dimensional space of observed spectra, and the samples projected onto these directions have the highest possibly variability. The features are represented by the eigenvectors associated with the *J* largest eigenvalues of the empirical covariance matrix of the training spectra. The centralized observed samples projected onto the space spanned by the orthogonal features determine the principal components (PCs). The variance of each PC is determined by the eigenvalue associated with the corresponding feature.

The scatter plots of each pair of three most significant PCs are illustrated in Figure 4 for the spectra of 13 bi-heterocyclic compounds (Figure 3). The notation of BHC1-BHC13 correspond to the chemical structures illustrated in Figure 2. The symbols “red plus” stand for the centroids of the clusters. Each class is represented by 30 samples.

PC1, PC2, and PC3 explain 61.03%, 32.21%, and 5.78% of the total variance. Thus, the explained variance by the first three PCs exceeds 99%. If J<<I, the reduction of dimensionality is very large. In our experiment, I=1194 and J=3. These results confirm that the THz spectra of bi-heterocyclic compounds contain highly redundant information, and a small number of PCs is sufficient to capture more than 99% of the total variance. However, three PCs are not sufficient to perform a successful classification, despite they capture a meaningful amount of variance. In general, the variance reflects the energy of a signal, and spiky signals, despite their high magnitude, do not carry high energy, but they are informative and might be discriminant. In other words, any spikes in a spectrum do not affect the variance considerably, but they may uniquely characterize a group of bi-heterocyclic compounds. Hence, we must use a higher number of PCs than comes from the variance analysis, which will be confirmed by the experiments. A similar conclusion was given in the paper [61] for classification of spectra in laser-induced breakdown spectroscopy (LIBS), and this behavior was also experimentally justified in the work [62].

### 3.2. KSVM

The samples shown in Figure 4 form consistent but non-Gaussian clusters that have arc structures, and some clusters cannot be separated with linear hyperplanes. This gives us empirical evidence that nonlinear hypersurfaces should be used to separate the groups of samples. To tackle this problem computationally, the observed samples can be projected onto the high-dimensional feature space using a nonlinear mapping $\Phi :R^{J}\to R^{D}$, where D>>J. The feature space can also be infinite dimensional Hilbert space of real-value functions with inner product <·,·> and norm ||·||. The aim is to find a nonlinear mapping Φ such that the samples $\left\{\Phi \left(z_{r}\right)\right\}$ are linearly separably in the feature space.

This is a basic concept of the nonlinear or Kernel Support Vector Machine (KSVM) that is successfully used in many areas of science [63,64]. The fundamental version of this classifier performs binary classification by finding the hyperplane $H=\{x:\;w^{T}\Phi \left(x\right)+b=0\}$ that best separates both classes in the feature space. The vector $w\in R^{D}$ is normal to H, and $\frac{b}{{\left||w|\right|}_{2}}$ is the perpendicular distance from H to the origin in $R^{P}$. The data points located closest to H are referred to as the support vectors (SVs). The best separating hyperplane should be selected in such a way that the distance between it and the point determined by any support vector in both classes is maximized.

Let $\left\{x_{r},y_{r}\right\}$ for r=1,…,R be a training dataset, where $y_{r}\in \left\{-1,1\right\}$ is an indicator of the class to which $x_{r}$ belongs. The task of finding the best separating hyperplane boils down to the constrained quadratic programming (QP) problem:
(1)$$\begin{array}{c} \min_{\lambda }\left(\frac{1}{2}\lambda^{T}H\lambda -e^{T}\lambda \right),\;s.t.\;y^{T}\lambda =0\;\mathrm{and}\;\forall r:C_{\xi }\geq \lambda_{r}\geq 0, \end{array}$$where $H=\left[h_{rs}\right]\in R^{R\times R}$ with $h_{rs}=y_{r}y_{s}\Phi^{T}\left(x_{r}\right)\Phi \left(x_{s}\right)$, $y=\left[y_{r}\right]\in R^{R}$, $e=\left[1,\ldots ,1\right]\in R^{R}$, and $C_{\xi }\geq 0$ is the soft-margin penalty parameter that is introduced to tackle the existence of inevitable outliers. The solution $\lambda_{*}$ to the optimization problem in formula (Equation 1) uniquely determines the parameters $w_{*}$ and $b_{*}$ in the separating hyperplane $H_{*}=\left\{x:w_{*}^{T}\Phi \left(x\right)+b_{*}\right\}$ that implies the decision rule for testing samples in *J*-dimensional space.

The mapping $\Phi (x_{r})$ transforms the samples $x_{r}$ from a *J*-dimensional to a very high-dimensional space. Fortunately, the inner product $\Phi^{T}\left(x_{r}\right)\Phi \left(x_{s}\right)$ does not have to be computed explicitly. Due to the Mercer’s theorem, the inner product $\Phi^{T}\left(x_{r}\right)\Phi \left(x_{s}\right)$ can be computed by the kernel function: (2)$$\begin{array}{c} k_{rs}=k\left(x_{r},x_{s}\right)=\Phi^{T}\left(x_{r}\right)\Phi \left(x_{s}\right), \end{array}$$where the matrix $K=\left[k_{rs}\right]\in R^{R\times R}$ is symmetric and semi positive-definite. Thus, the QP problem in expressed by (Equation 1) is convex and can be easily solved by many solvers using, e.g., the active set or interior point algorithm. In the experiments, we used the Sequential Minimal Optimization (SMO) algorithm.

The kernel function in (Equation 2) can take various forms. For example:***d*-degree polynomial:**$k_{1}\left(z_{r},z_{s}\right)={\left(z_{r}^{T}z_{s}+\beta_{0}\right)}^{d}$, where d≥1 and $\beta_{0}\geq 0$,**Gaussian Radial Basis Functions (GRBF):**$k_{2}\left(z_{r},z_{s}\right)=\exp\left\{\frac{\left|\right|z_{r}-z_{s}{\left|\right|}^{2}}{2\sigma^{2}}\right\}$, where $\sigma^{2}>0,$**two-layer perceptron:**$k_{3}\left(z_{r},z_{s}\right)=\tanh\left(\beta_{0}z_{r}^{T}z_{s}+\beta_{1}\right)$, where $\beta_{0},\beta_{1}>0$.

Note that, if d=1 and $\beta_{0}=0$ in $k_{1}\left(z_{r},z_{s}\right)$, then KSVM with this kernel simplifies to the standard SVM. When d=2 and J=2, i.e., $z_{r}={\left[z_{1r},\;z_{2r}\right]}^{T}$ and $z_{s}={\left[z_{1s},\;z_{2s}\right]}^{T}$, we have $\Phi \left(z_{r}\right)={\left[z_{1r}^{2},z_{2r}^{2},\sqrt{2}z_{1r}z_{2r},\sqrt{\beta_{0}}z_{1r}z_{2r},\sqrt{2\beta_{0}}z_{1r},\sqrt{2\beta_{0}}z_{2r},\beta_{0}\right]}^{T}\in R^{7}$. For a larger value of *J*, the dimensionality of the feature space is much bigger even for a second-degree polynomial. Fortunately, due to the kernel assumption, the mapping Φ is never computed explicitly in the whole KSVM algorithm.

The curvature of the nonlinear separating hypersurface can be controlled by the kernel function and its related parameters. It is obvious that the curvature should be adapted to data, which implies that the kernel parameters are learnt from training data. Let $p=\{d,\beta_{0},\beta_{1},\sigma^{2},C_{\xi }\}$ be a vector of the parameters to be estimated. To estimate the vector ***p***, the *n*-fold CV was applied separately to each class.

The main criterion used for testing and validating the model is the Misclassification Ratio (MCR) that is defined as $\Psi =\{♯t:y_{t}-y_{t}^{\left(true\right)}\neq 0\}$, where $y_{t}$ is a binary variable expressed by the decision rule in KSVM, and $y_{t}^{\left(true\right)}$ is the true label of the class to which the *t*-th testing sample belongs to. Note that the function Ψ is parameter-dependent. Thus, the problem of data-driven estimation of the parameters can be formulated by the following constrained optimization problem: (3)$$\begin{array}{c} \min_{z}\Psi \left(z\right),\;s.t.\;\forall l:c_{l}\left(p\right)\geq 0, \end{array}$$where $c_{l}\left(z\right)$ are constraints.

The function Ψ(z) is not differentiable with respect to ***z***. To solve problem (Equation 3), derivative-free methods must be used, such as evolutionary strategies, metaheuristics, or direct search methods [65,66,67]. In our experiments, we chose the Nelder–Mead (NM) method [68] that belongs to a family of direct search strategies for multi-variate unconstrained optimization. It generates a sequence of simplexes for which the objective function at their vertices decreases its value. Starting from (J+1)-dimensional simplex in the space $R^{J}$, the vertices of the current simplex are updated according to an expansion–shrink strategy that searches for reflection–contraction points that are used in decision rules to expand or shrink the current simplex. The NM method is basically used for unconstrained nonlinear optimization, but it can be applied to problem (Equation 3) because the constraints in all of the above-mentioned kernel functions are strictly positive and can be expressed by the following exponential mapping: $z=\exp\{\tilde{z}\}$, where $\tilde{z}$ is an unconstrained vector of search variables in the NM method.

The standard KSVM is a binary classifier. Thus, it can be directly applied to binary decision problems, e.g., to decide whether a given THz spectrum represents a given bi-heterocyclic compound or not. For multiple bi-heterocyclic compounds, one-vs.-all classification strategy can be applied. Hence, we use as many KSVM classifiers as there are classes, and each classifier is trained to recognize one bi-heterocyclic compound against the rest. Then, in the testing process, a testing sample is verified separately by each classifier. Note that this methodology incurs a high computational cost, but it offers many additional advantages. For example, if a testing sample cannot be identified by any trained classifier, we can apply another classifier only to the unrecognized sample, or we can assign this sample to some unknown class. The samples that fall to such a new class are candidates for representing a new bi-heterocyclic compound, and they are further classified with the *k*-NN classifier. Similarly, if a testing sample is recognized by more than one classifier, we can also repeat the classification with another, more efficient, classifier. This approach is particularly useful in practice, when outliers or other perturbations occur.

## 4. Classification Results

Various experimental tests are performed to demonstrate efficiency of the proposed approach in bi-heterocyclic compounds (BHC) recognition. In the first stage, the KSVM with an additional class and the non-adaptive GRBF kernel was used, where $\sigma^{2}=C_{\xi }=1$. In the second stage, the parameter $\sigma^{2}$ was learnt automatically from the observed spectra. In the preliminary tests, we observed that the soft-margin $C_{\xi }$ has a negligible effect on the accuracy of recognition, hence we assumed $C_{\xi }=1$ in the further tests. Neglecting the updates for the variables ***w*** and *b* in the primal QP problem, problem (Equation 3) boils down to one variable nonlinear optimization problem.

In the experiments, we used 390 absorbance spectra registered by the THz-TDS spectrometer for 13 bi-heterocyclic compounds that are described in Section 2.3. The spectral resolution amounts to 1194 subbands, and each compound is represented by 30 spectra. All of the originally registered spectra were preprocessed with PCA where the number of PCs was selected in the range (1,40). The training was performed on 12 selected compounds labeled according to Figure 2. The 13th compound is regarded as unknown, and its spectra will be referred to as outer-class samples which will be used only for testing. Hence, they form an additional (extra) class (labeled as 13). Moreover, the testing samples are not taken from the same spectra dataset that is used for training. Due to the long-term drift effects and other slowly-varying conditions of measurements, the spectra used for testing are measured several days later than the other samples. The testing set consists of 30 samples of mixed classes, including the outer-class. We analyze 11 scenarios with an unknown compound, changing the number of spectra from the outer-class in the testing dataset, from zero to 100% with the step of 10%. To check the repeatability of the results, each scenario was repeated 13 times—each time one compound was neglected. It means that the training process was performed for each scenario with one (each time different) missing class, and, in the testing, the samples from the missing class should be classified to the extra class.

Since the objective function in (Equation 3) is non-convex with respect to ***z***, the NM method may fall in local minima. To minimize such a risk, multi-start initialization is necessary. Thus, each experiment with data-driven parameter estimation was also repeated with respect to the multi-start NM initialization. Finally, the MCR is averaged separately for each scenario.

The proposed algorithm was coded in MATLAB R2016a and run on the machine equipped with two CPU Intel Xeon(R) E5-2699V3 (Santa Clara, CA, USA), 2.30 GHz.

### 4.1. Untrained GRBF Kernel

The averaged accuracy of BHC recognition obtained by the KSVM with the untrained GRBF (fixed $\sigma^{2}=1$) versus the number of PCs is shown in Figure 5. This algorithm yields the mean accuracy of recognition greater or equal to 90% only for the PCs in the range [10,21]. The accuracy exceeds 97% for three values of PCs (J=13,18,19) with the highest rate equal to $\eta_{avg}=97.93\%$ for J=18. For other numbers of PCs in the range [1,40], the accuracy is below 90%.

Figure 6 presents the mean accuracy versus PCs in the range [13,20] (non-blue bars in Figure 5) at various percentage levels of contribution of the outer-class bi-heterocyclic compound (from zero to 100% at the step of 10%). The accuracy changes considerably with the number of PCs and the number of outer-class samples. The accuracy reaches 100% for J=13, when all the test samples are known, i.e., they belong to the classes that are used for training. If outer-class samples occur and a low value of PCs is used, the efficiency of recognition falls down noticeably. The accuracy averaged over the contribution of outer-class samples is about 95.13%. For J∈[20,40], we observed a reverse behaviour. When all the testing samples belong to an extra class, $\eta_{avg}=100\%$, while, if some testing samples from other classes occur, the accuracy drops considerably. For J=20 and all the known samples, $\eta_{avg}=93.59\%$, while we have only $\eta_{avg}=18.46\%$ for J=40. For J=13 and J=18, the accuracy was similar, i.e., 97.88% and 97.93%, respectively. Thus, the choice of the parameter *J* may be conditioned by our expectation—better recognition of unknown compounds or better classification of testing samples that are closely related to training samples. The optimal choice seems to point at J=18 with the worst accuracy of 96.92%, and the best—99.23% if all the testing samples represent a new compound. When J=18, the explained variance amounts to 99.9984%, but the MCR at the level of 3% (in the worst case) and its variability with the number of PCs in quite a narrow range is not fully satisfactory in applications to pharmaceutical industry and medal chemistry.

The mean training time for the discussed case (J=18) is 391.8 ms with the standard deviation (STD) of 3 μs. The testing time obviously depends linearly on the number of testing samples, and can be determined by the regression line $ET\left(n_{t}\right)=3.9n_{t}+1.4$ [ms] with the confidence interval of 95%, where $n_{t}$ is the number of testing samples. The overall time for testing 30 samples takes about 120±1.3 ms.

### 4.2. Trained GRBF Kernel

The results obtained with the adaptive KSVM are shown in Figure 7 versus the number of PCs. For this approach, we were able to obtain 100% mean accuracy of BHC recognition for J≥34. The tests are performed for the 2-fold CV for training the kernel parameters and 30 initialization points in the NM algorithm. The proposed algorithm is resistant to the amount of outer-class samples in the testing set. Reducing the dimensionality from 1194 subbands to 34 PCs, 99,9957% of the total variance is preserved.

For J∈[1,21], the mean accuracy does not exceed 90 % but when J≥30, the advantage of the KSVM with the trained GRBF is considerable with respect to the untrained case. When the testing samples contain a new BHC, the KSVM with the trained GRBF can adapt to each analyzed case when J≥25, and is able to correctly classify all the testing samples, if they do not contain outer-class samples—see the results presented in Figure 8. Moreover, in this respect, it outperforms the KSVM with the untrained GRBF considerably—compare the results presented in Figure 6 and Figure 8.

In the range J∈[30,40], the accuracy is nearly constant with CV folds used for training the kernel. However, the number of CV folds has a great impact on the training time, which is illustrated in Figure 9. The results are obtained for J=35 and 30 initialization points in the NM algorithm, where 99.9959% of the total variance is explained. The median of training time of KSVM combined with running PCA changes from 6.6822 s. (2-fold CV) to 26.2272 s. (10-fold CV).

We also analyzed how an increase in the number of CV folds affects a minimal number of PCs needed to reach the accuracy of 100%. For the CV with 5–10 folds, 100% accuracy is possible at J=33. The time profit obtained from a decrease *J* by two at 5-fold CV with respect to J=35 at the 2-fold CV does not compensate the total time lost of 7.1333 s. Another parameter that considerably affects the accuracy and runtime is the number of initialization points in the NM algorithm. Figure 10 presents the accuracy vs. the parameter *J* at various numbers of initialization points [5,10,15,20,30,50].

If the number of these points is too small, there is a higher probability that the solution to the problem (Equation 3) is suboptimal. Figure 10 shows that the accuracy of 100% is not obtained in the whole range of PCs (J∈[30,40]) when only five points are used. However, when 10 points are used, all of the test samples are correctly recognized at J=34. When the number of the points exceeds 30, the highest accuracy is observed in the desired range of PCs.

The training time of the KSVM with an adaptive GRBF versus the number of initialization points at J=35 and 2-folds CV is listed in Table 1.

The training time increases exponentially with the number of initialization points in the NM algorithm, and it can be modelled by the regression curve (with a confidence level of 95%): $ET_{\left(train\right)}\left(s\right)=-2.371\exp\left\{-83.45\times 10^{-3}s\right\}+5.349\exp\left\{8.371\times 10^{-3}s\right\}$, [s], where *s* is the number of the points. Considering the obtained accuracy, training time, and non-convexity of the NM algorithm, the optimal number of the initialization points seems to be 30.

For the KSVM with the trained GRBF with J=35, the mean training time amounts to 6.71 ± 0.18 s, and its median is 6.68 s. Similarly as for the untrained kernel, the testing time can be approximated by a linear function $ET\left(n_{t}\right)=0.369n_{t}+0.536$ [ms] with the confidence interval of 95%, where $n_{t}$ is the number of testing samples. The overall time for testing 30 samples takes about 11.6ms.

## 5. Conclusions

The method for classification of bi-heterocyclic compounds developed by us can be successfully used in medical chemistry and pharmaceutical industry. The compound spectra (BHC1-13) recorded with the THz-TDS terahertz spectrometer provide enough information, so that the compounds can be classified unambiguously by means of machine learning methods. It should be remembered that the compounds selected for the study have the same core (isothiazolpyridine derivatives) in the chemical structure. Assigning the spectrum of an unknown sample to a specific group can be performed with many classification methods. However, the KSVM trained separately to each class gives us a possibility to assign a testing sample to an unknown group which can be analyzed with other methods, e.g., with the *k*-NN classifier.

In medical chemistry, the presence of new compounds is very desirable and it is important to determine quickly and unequivocally whether it is a new structure or a known, previously synthesised compound. The KSVM with the untrained kernel does not meet our expectations because, depending on the percentage of unknown content of samples in the test set, for the best case (for 18 PCs), allows for one to four misclassifications per every 100 samples.

This problem was solved by using the KSVM with the trained GRBF. The proposed algorithm assumes the splitting of the classification problem of *C* known compounds into *C* binary problems, which allows the creation of an additional (extra) class C+1 to which a new, previously unknown sample can be assigned. In this way, using e.g., cross-validation and and the derivative-free method for training the kernel, the algorithm adjusts the parameters on its own (***w***, *b* and the kernel scaling function σ) optimizing the hyperplane separating individual classes. After creating a single decision rule using samples from cross-validation, the test samples of any bi-heterocyclic compounds are classified. When carrying out the testing *C* times, three results are obtained. The sample can be assigned an index of a known class, an index of an unknown class (C+1) or several indexes of known classes. In the third case, the final classification decision is taken with the help of *k*-NN with the Euclidean metric. For the studied group of BHC1-13 compounds, we achieved the efficiency of classification η = 100% regardless of the number of unknown bi-heterocyclic compound samples in the test set. The optimal solution is provided by the use of PCA pre-processing with 35 components, 2-fold cross-validation and 30 starting points in the NM algorithm. The training time of our algorithm is much longer than for the KSVM with the untrained Gaussian kernel (6.7 s vs. 390 ms), but may be partially compensated by the time of sample recognition, especially when loading long series of spectra. In the range of 1–30 test samples, we gain the time spent on sample classification from about 6 (comparison for 1 sample in the test set: 4.675 ms vs. 0.792 ms) to more than 10 times (comparison for 30 samples: 120 ms vs. 11.6 ms).

In summary, the results show that the proposed THz-TDS system can perfectly classify all the BHCs, regardless of the number of outer-class samples in the testing set. In this study, we focused on classification of the BHCs having the common core and different molecular tails (linkers with functional groups) that inhibit cyclooxygenases COX-1 and COX-2. Taking into account two opposite measurement cases with the testing samples—all belonging to the space of training samples or all are outer-class samples—we can conclude that, in the former case, it is possible to recognize all molecular tails linked to the same core. In the other case, we observed that the proposed sensor system can correctly recognize new tails and classify them to a new group, even if their structures are similar to those used for training (e.g., BHC1 and BHC2 in Figure 1). It is therefore the method sensitive to any changes introduced to the BHC. The analysis of BHCs with other cores will be performed in our future research.

## Figures and Tables

**Figure 1 sensors-19-03349-f001:**
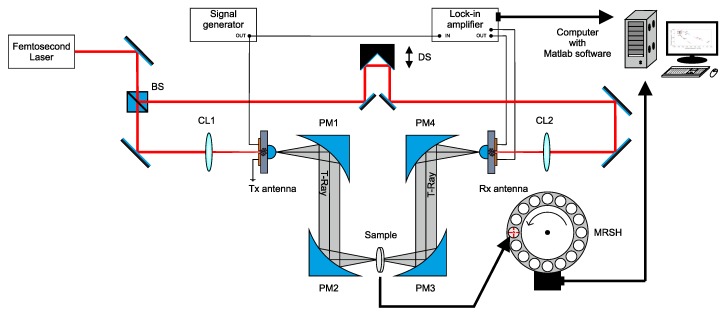
Terahertz system for the recognition of bi-heterocyclic compounds.

**Figure 2 sensors-19-03349-f002:**
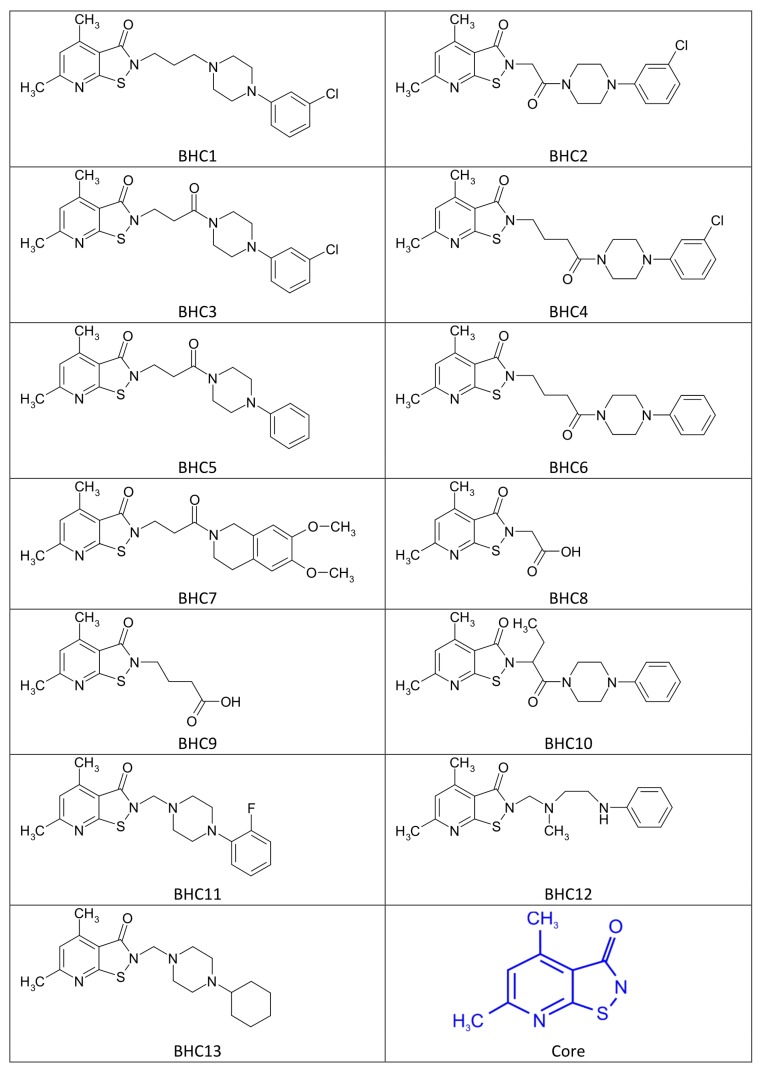
Chemical structures of bi-heterocyclic compounds (BHC1-13) used in the studies. The common core is marked in blue.

**Figure 3 sensors-19-03349-f003:**
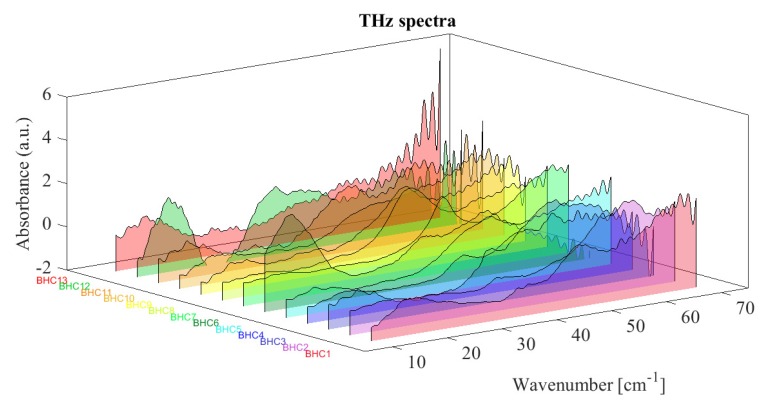
THz spectra of tested bi-heterocyclic compounds.

**Figure 4 sensors-19-03349-f004:**
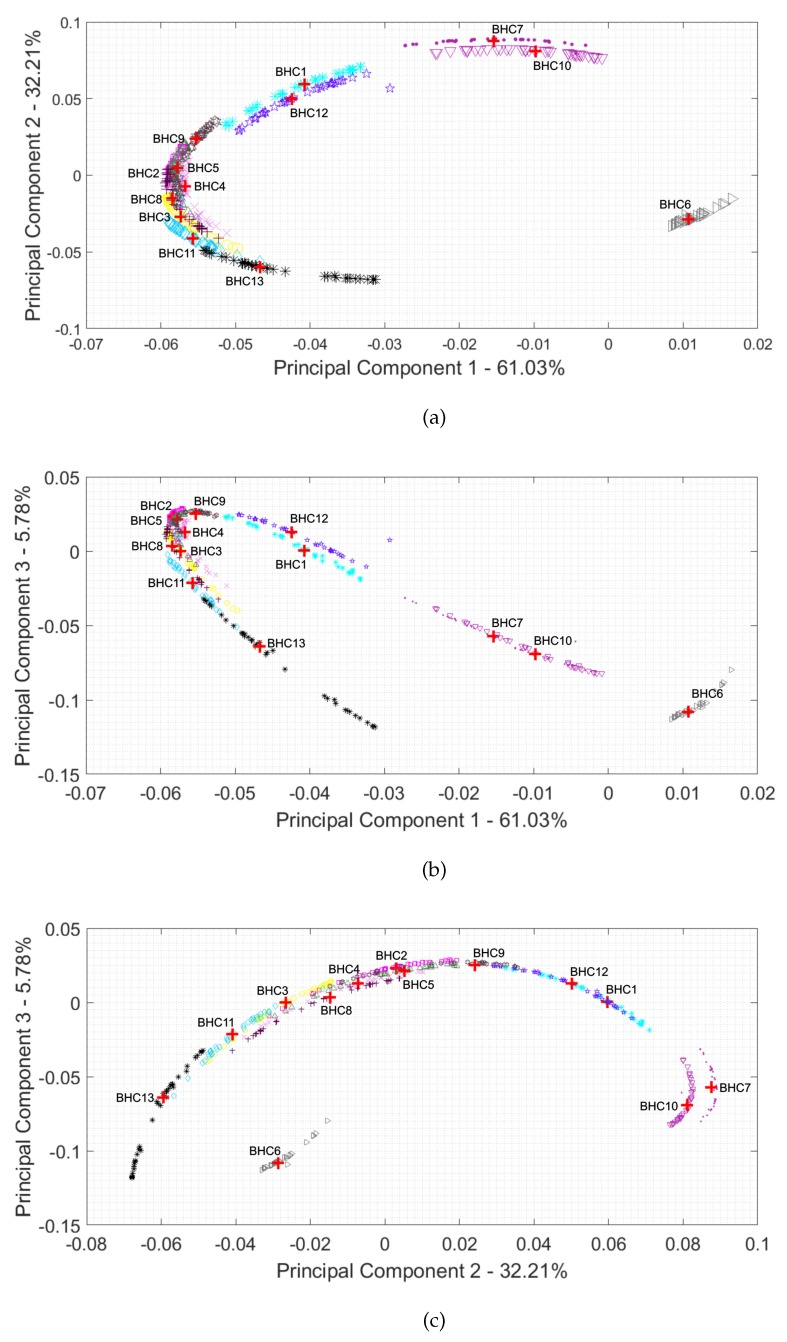
PCA map of thirteen bi-heterocyclic compounds: (**a**) PC1 and PC2; (**b**) PC1 and PC3; (**c**) PC2 and PC3.

**Figure 5 sensors-19-03349-f005:**
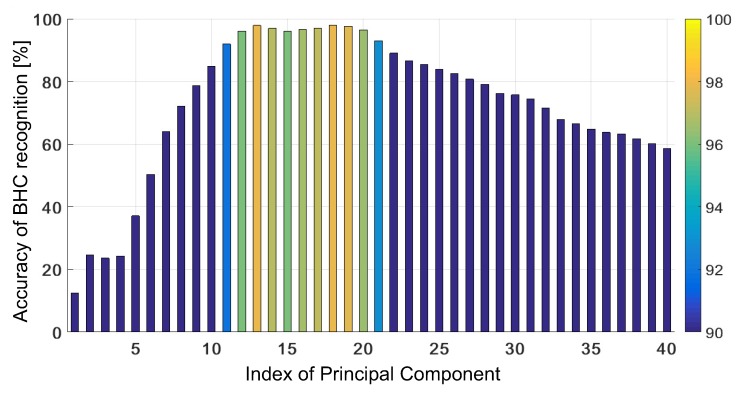
Accuracy of BHC recognition (average) versus PCs for the KSVM with the untrained GRBF.

**Figure 6 sensors-19-03349-f006:**
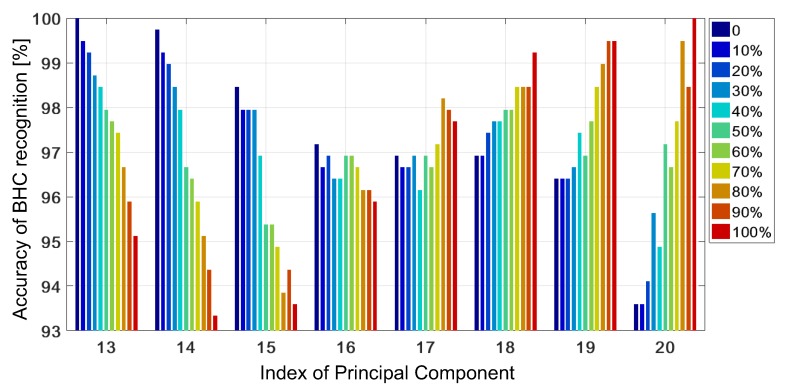
Accuracy of BHC recognition obtained with non-adaptive KSVM versus the number of PCs (from 13 to 20) at various contributions of outer-class samples in the test set.

**Figure 7 sensors-19-03349-f007:**
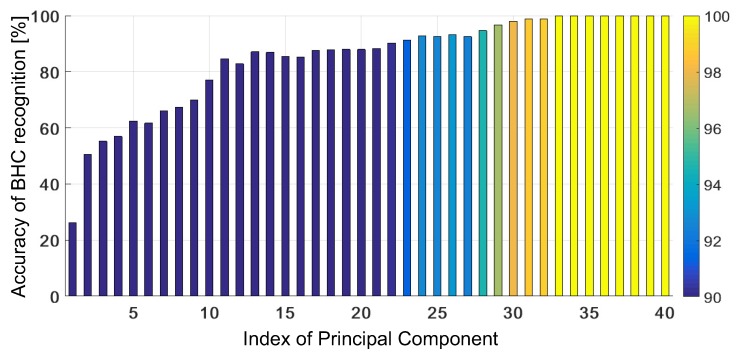
Accuracy of BHC recognition (average) obtained with adaptive KSVM versus the number of PCs. The test was performed using 2-fold CV and 30 initialization points for the NM algorithm.

**Figure 8 sensors-19-03349-f008:**
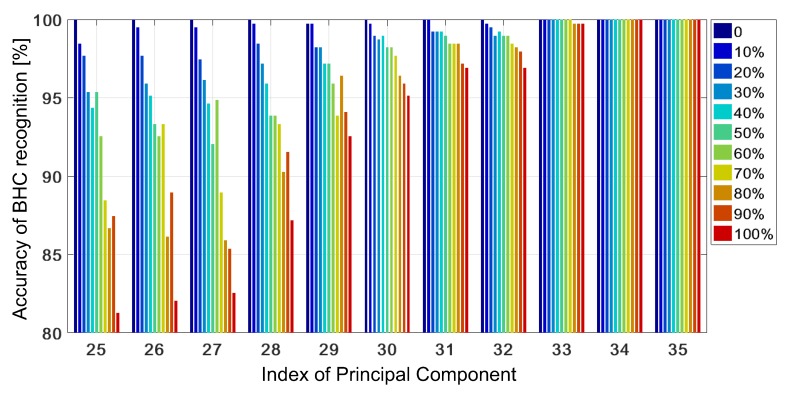
Accuracy of BHC recognition obtained with adaptive KSVM versus the number of PCs (from 25 to 35) at various contributions of outer-class samples in the test set.

**Figure 9 sensors-19-03349-f009:**
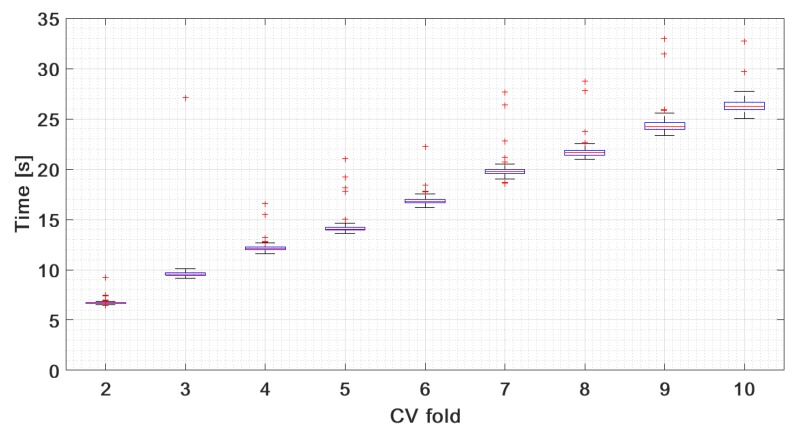
Time-training statistics (box plots) obtained for the adaptive KSVM versus the number of CV folds. The test was performed at 35 PCs and 30 initialization points in the NM algorithm.

**Figure 10 sensors-19-03349-f010:**
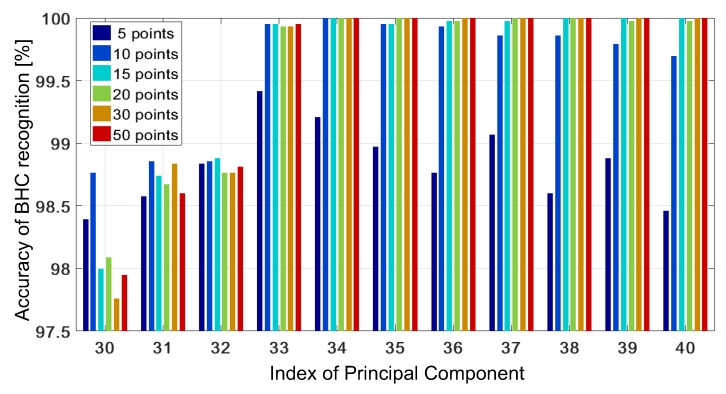
The accuracy of BHC recognition versus the number of PCs for various numbers of initialization points in the NM algorithm.

**Table 1 sensors-19-03349-t001:** The mean training time [s] of the KSVM with an adaptive GRBF versus the number of initialization points in the NM algorithm.

No. initialization points:	5	10	15	20	30	50
Mean training time [s]:	4.01	4.84	5.34	5.97	6.7	8.22

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
