# Peer review of "Recognition of Pharmacological Bi-Heterocyclic Compounds by Using Terahertz Time Domain Spectroscopy and Chemometrics"

_sensors, 2019, doi:10.3390/s19153349_

Round 1
Reviewer 1 Report
The basis of this research paper is supposed to investigate classification and also recognition of several pharmacological bi-heterocyclic compounds by terahertz time-domain spectroscopy combining with chemometrics. The subject addressed in the manuscript is, in my opinion, interesting. However, the quality of this manuscript could not meet the journal desired standard of Sensors, and it should not be accepted at this stage. 1. This submission should belong to be a research paper meanwhile it looks like a review work with too many unrelated description context in both introduction section and also statistical analysis section. In the introduction section, they mentioned “1.1. Drug development, 1.2. Time and costs….” Such context are actually no any meaningful thinking for the focus point about classification and also recognition of several pharmacological bi-heterocyclic compounds by using spectroscopy combining with chemometrics. Such as the following description about drug development “In addition, less than 1/100,000 compounds obtained in the course of research work appear on the pharmaceutical market [25], which indicates a multitude of test studies aimed at selecting the best drug. The review of relevant literature indicates that research is becoming more and more expensive - at the turn of the 20th and 21th century the average amount spent on the dev elopment of a new drug was much lower and amounted to about 800 million US dollars [26,27]”……. Also in the following statistical analysis section, there are too much bored descriptions in 3.1. PCA and 3.2. KSVM, which could be found in even any text books. 2. There are no real discussions about classification results of several pharmacological bi-heterocyclic compounds based on terahertz spectra combining with chemometrics. Actually the authors did not mention why they would use the spectral results shown in terahertz region, probably NIR, mid-IR or even Raman spectra could also be even better in classification and recognition of such bi-heterocyclic compounds combining with chemometrics? In abstract the authors mentioned that “Our approach allows us to achieve 100% accuracy of the classification of the tested chemical compounds in the time of several milliseconds counted for 30 samples of the test set.”, actually 100% accuracy could be possible? (even they suggested that how an increase in the number of CV folds affects a minimal number of PCs needed to reach the accuracy of 100%. And 100% accuracy is possible at J = 33??) . 3. There are several studies on the use of SVM and terahertz spectra for the prediction of physicochemical parameters or identification of drugs other than prohibited. Such as references [72. Sterczewski, L.A.; Nowak, K.; Szlachetko, B.; Grzelczak, M.P.; Szczesniak-Siega, B.; Plinska, S.; Malinka, W.; Plinski, E.F. Chemometric evaluation of THz spectral similarity for the selection of early drug candidates. Scientific reports 2017, 7, 14583. and 76. Jingling, Z.S.C.S.S.? Identification of Terahertz Absorption Spectra of Illicit Drugs Using Support Vector Machines [J]. Chinese Journal of Lasers 2009, 3.], any relationship or difference should be given about this work comparing with previous work? 4. Again the number of references and unrelated references shown in this manuscript is too tediously long, and also the reference style is not unified and even totally wrong (such as 18. Das, P.S.; Saha, P. A REVIEW ON COMPUTER AIDED DRUG DESIGN IN DRUG DISCOVERY 2017.???no any volume or page? 27. Borchardt, J. The Alchemist 6 December 2001.???? 73. Chen, Y.; Liu, Y.; Zhao, G.; Wang, W.; Li, F. Chinese traditional medicine recognition by support vector machine (SVM) terahertz spectrum. Guang pu xue yu guang pu fen xi= Guang pu 2009, 29, 2346–2350……..). The authors should check every of them carefully. 5. This work seems not related with the science and technology of sensors and biosensors which is the scope of Sensors journal. After the major revision (such as shorten the context and references…..), it is recommended to be submitted in the following journals about spectroscopy (such as Journal of Applied Spectroscopy, Applied Spectroscopy, Spectroscopy, IEEE Transactions on Terahertz Science and Technology).Author Response
Good afternoon. In the attachment I send our answers.

Reviewer 2 Report
The work is an implementation of a system to detect bi-heterocyclic compounds. The authors employed THz spectrum with KSVM.
For the recognition the authors achieved a 100% accuracy. It is good that the authors had 30 samples to test which I personally see is more than adequate.
While detecting it is important to determine quickly irrespective it is a new chemical structure or not. The employed method can be cost effective and hopefully time effective.
The paper itself is flawless and I did not see even any single spelling/typo error which I rarely see while reviewing other papers. Not even in the reference. I hope it can be published without any major or minor correction.
Author Response
Good afternoon. In the attachment I send our answers.

Round 2
Reviewer 1 Report
I am happy with the responding letter and revised manuscript. As the authors have answered all questions and assured that the main contribution of the manuscript falls in an application area of the machine learning methods, which should be for statistical interpretation of the signals provided by a specific sensor – the THz-TDS system.
The quality of this revision has been improved a lot so I recommend this revised work for publication in Sensors.